# Robust Multi-Agent Pathfinding with Continuous Time

## Abstract

Multi-Agent Pathfinding (MAPF) is the problem of finding plans for multiple agents such that every agent moves from its start location to its goal location without collisions. If unexpected events delay some agents during plan execution, it may not be possible for the agents to continue following their plans without causing any collision. We define and solve a $T$-robust MAPF problem that seeks plans that can be followed even if some delays occur, under the generalized MAPF$_R$ setting with continuous time notions. The proposed approach is complete and provides provably optimal solutions. We also develop an exact method for collision detection among agents that can be delayed. We experimentally evaluate our proposed approach in terms of efficiency and plan cost.

## Introduction

Multi-Agent Pathfinding (MAPF) is the problem of finding plans for multiple agents such that every agent moves from its start location to its goal location without any collision. MAPF can be applied to autonomous vehicles (Morris et al. 2016), robotics (Veloso et al. 2015), video games (Ma et al. 2017), warehouse management (Wurman, D'Andrea, and Mountz 2008), etc. Solving MAPF optimally for common objective functions is NP-hard (Yu and LaValle 2013a; Surynek 2010). Nonetheless, practically efficient algorithms have been developed (Wagner and Choset 2015; Surynek 2012; Boyarski et al. 2015; Yu and LaValle 2013b).

Most prior works on MAPF assumed that time is discretized into time steps and each action executes in one time step (Andreychuk et al. 2022). However, these simplified assumptions restrict the applicability of MAPF algorithms in real-world scenarios. In addition, unexpected events may delay some of the agents during the execution of a MAPF plan, preventing them from following their plans on time. In this case, we may need to replan for the agents to avoid collisions. Such re-planning may not always be practical, as it requires resources (e.g., computing and communication), or time that the agents may not have. Thus, it is often desirable to generate plans that can tolerate unexpected delays.

In this paper, we explore a continuous-time form of robustness for MAPF called $T$-robust MAPF ($T$R-MAPF), which is designed to generate MAPF plans for continuous-time scenarios that can be followed even if unforeseen delays occur. In $T$R-MAPF, we seek a plan that is robust to

a total time delay of $T$ per agent during plan execution, i.e., each agent may be delayed up to $T$ and the plan remains safe (collision-free). In the continuous-time form, the time delay $T$ can be any non-negative real number.

Our contributions can be summarized as follows:

- $T$**-Robust Continuous-time Conflict-based Search ($T$R-CCBS)**: We present sufficient and required conditions for a MAPF solution to be $T$-robust and develop an extension to CCBS to find $T$R-MAPF plans in a state space that grows exponentially with $T$.

- **Closed-form formulas for Collision Detection under $T$-delay**: We propose a collision detection mechanism under $T$-delay, which is also used to compute unsafe intervals under $T$-delay. To improve accuracy and reduce computational complexity, closed-form formulas for collision detection under $T$-delay are developed.

The rest of the paper is organized as follows: First, we summarize some closely related work. Next, $T$R-MAPF is introduced, and we propose our $T$R-CCBS approach to solve $T$R-MAPF. After that, the closed-form formulation for collision detection under $T$-delay is presented. Finally, the paper finishes with empirical evaluation and conclusions.

## Related Work

Ma, Kumar, and Koenig (2017) proposed a MAPF algorithm based on conflict-based search (CBS) (Sharon et al. 2015) to minimize the expected makespan as part of their work on execution policies for MAPF with delays. However, they assumed prior knowledge of delay probabilities and did not provide any robustness guarantee on the plan constructed.

Atzmon et al. (2018) proposed $k$-robust CBS that can find $k$-robust plans, where each agent can tolerate up to $k$ time steps of delay while executing the plan without collisions. They also extended the work to develop several robust execution policies guaranteeing that the agents reach their goals even if they encounter unexpected delays (Atzmon et al. 2020). However, they focused on discrete time notions with unit time steps and did not support continuous time notions.

Walker, Sturtevant, and Felner (2018) introduced a MAPF formulation with non-unit duration actions and extended the increasing cost tree search algorithm for it. Cohen et al. (2019) enhanced CBS to reason with continuous time and developed a bounded-suboptimal extension of Safe Interval

Path Planning (SIPP). Andreychuk et al. (2022) proposed continuous-time conflict-based search (CCBS), a MAPF algorithm that handles continuous time notions and is sound, complete, and optimal. They also adapted successful CBS improvements, including prioritizing conflicts, disjoint splitting, and high-level heuristics, to CCBS (Andreychuk et al. 2021). However, all the above algorithms do not support robustness or tolerate any delays experienced by the agents.

## $T$-Robust MAPF

We consider MAPF in 2D environments. For ease of exposition, all agents are assumed to have the same (1) shape and size, and (2) constant speed. The environment is defined as a graph $G = (V, E)$ whose vertices correspond to locations that agents can occupy (and wait in them) and edges correspond to straight-line trajectories for agents to move from one location to the other. We ignore the inertia and assume that the agents start or stop moving instantaneously. The duration of a move action is the length of the edge divided by the speed, while the duration of a wait action can be any positive real number. Prior work referred to this setting as $\text{MAPF}_R$ (Walker, Sturtevant, and Felner 2018).

There are $n$ agents. Each agent $i$ has a start vertex $s_i \in V$ and a goal vertex $g_i \in V$. A plan $\pi_i$ for an agent $i$ is a sequence of actions that moves the agent from its start vertex to its goal vertex. Each action in the plan is a pair $(a_i, t_i)$ representing that action $a_i$ (either move or wait) starts execution at time $t_i$. The end time of an action is the start time of the next action in the sequence. After finishing the plan, the agent stays at the last location (its goal vertex) forever.

A set of plans, one for each agent, is called a joint plan, denoted by $\pi = \{\pi_1, ..., \pi_n\}$. A solution to a $\text{MAPF}_R$ problem is a joint plan such that if all agents start to execute their plans concurrently, they will reach their goal vertices without any collision. The cost of a joint plan is defined by an objective function. Common objective functions include makespan and sum of costs, which refer to the maximum and total times for agents to reach their goal vertices.

A conflict is defined between two timed actions.

**Definition 1** (Conflict). A conflict is a tuple $\langle a_i, t_i, a_j, t_j \rangle$, such that if agent $i$ executes action $a_i$ at time $t_i$ and agent $j$ executes action $a_j$ at time $t_j$, they will collide (i.e., their geometry shapes will overlap at some time during the actions).

A joint plan $\pi$ is said *valid* if it is conflict-free. A MAPF solver is *sound* if it outputs a valid plan. A *delay* in a plan $\pi$ is defined by a tuple $\langle i, t_D, \Delta_D \rangle$, representing that agent $i$ experiences a delay $\Delta_D$ starting from time $t_D$. This means that for each action $(a_i, t_i)$ after $t_D$ ($t_i \geq t_D$) in the plan, agent $i$ will not perform action $a_i$ at the intended time $t_i$ and instead will perform $a_i$ at time $t_i + \Delta_D$. A plan is *robust* to a delay if the delayed agent can continue to execute its remaining plan after the delay without causing any collision.

Formally, for a plan $\pi$ and a delay $D = \langle i, t_D, \Delta_D \rangle$, let $D(\pi)$ be the plan that is equivalent to $\pi$, except for replacing each timed action $(a_i, t_i)$ in $\pi_i$ with $(a_i, t_i + \Delta_D)$ if $t_i \geq t_D$. A plan $\pi$ is robust to a delay $D$ if $D(\pi)$ is valid. $\pi$ is robust to a series of delays $\mathcal{D}$ if applying all the delays in $\mathcal{D}$ to $\pi$ will yield a valid plan, i.e., assuming

$\mathcal{D} = \{D_1, D_2, \ldots, D_m\}$ where $t_{D_1} \leq t_{D_2} \leq \cdots \leq t_{D_m}$, the plan $D_m(\ldots D_2(D_1(\pi))\ldots)$ is valid. Note that there could be multiple delays in $\mathcal{D}$ on the same agent.

**Definition 2** ($T$-robust Plan). A plan is $T$-robust if it is valid and it is robust to any series of delays that contains at most a total delay of $T$ for each agent.

We extend the definition of a conflict between two timed actions to a $T$-delay conflict to facilitate checking if a plan is $T$-robust.

**Definition 3** ($T$-delay Conflict). A $T$-delay conflict is a tuple $\langle a_i, t_i, a_j, t_j \rangle$, such that there exists a value $\Delta \in [0, T]$ so that if agent $i$ executes action $a_i$ at time $t_i$ and agent $j$ executes action $a_j$ at time $t_j + \Delta$, they will collide.

**Lemma 1.** A plan is $T$-robust if and only if it does not contain any $T$-delay conflict.

Due to space limitations, we omit the formal proof of Lemma 1. There can be more than one $T$-robust plan for a given $\text{MAPF}_R$ problem. A $T$-robust plan is *optimal* if there is no other $T$-robust plan with a lower cost.

## $T$-Robust Continuous-time CBS ($T$R-CCBS)

### Review of CBS

CBS (Sharon et al. 2015) is a widely used MAPF solver which is complete and optimal. It is designed for classical MAPF with discrete time notions and unit-duration actions. It solves a given MAPF problem by building a plan for each agent separately, detecting conflicts between these plans, and resolving them by replanning for involved agents with additional constraints that they cannot occupy certain vertices or traverse certain edges at particular time steps.

CBS searches in a Constraint Tree (CT) for a set of constraints that an optimal plan should satisfy. The CT is a binary tree, where each node $N$ represents a set of constraints imposed on the agents ($N.constraints$) and a joint plan satisfying these constraints ($N.\pi$). The root node has an empty set of constraints. A successor node inherits the constraints of the parent node and adds a new constraint for one agent.

Next, we describe how CBS identifies conflicts in a node $N$ and chooses the constraint to add when expanding $N$ and generating its successors.

**Identifying conflicts** For each node $N$, a low-level solver is used to find a plan $N.\pi$ for each agent, subject to the constraints for the agent in $N.constraints$. The plan is validated by simulating the movement of the agents along their planned paths. $N$ is a goal node if $N.\pi$ does not contain any conflict. $N$ is a non-goal node if a conflict is found in $N.\pi$.

**Resolving conflicts** CBS performs a best-first search of the CT. In each iteration, the CT node with the lowest-cost joint plan is selected. When a goal node $N$ is selected, CBS stops searching and outputs its plan $N.\pi$. When a non-goal node $N$ is selected, CBS chooses a conflict in $N.\pi$ and resolves it by generating two successor nodes of $N$. Specifically, if the conflict chosen involves two agents $i$ and $j$ at $x$ (either a vertex or an edge) at time step $t$, two successor nodes $N_i$ and $N_j$ are generated by inheriting $N.constraints$

and adding a new constraint that prohibits agents $i$ and $j$ respectively from occupying or traversing $x$ at $t$. After that, CBS continues searching the CT in a best-first manner.

## $T$-Robust CCBS

Next, we introduce $T$-robust CCBS ($T$R-CCBS), an adaptation of CCBS designed to return optimal $T$-robust plans.

CCBS (Andreychuk et al. 2022) extends CBS for MAPF$_R$ by using a geometry-aware collision detection mechanism. To avoid collisions, CCBS computes constraints according to the *unsafe intervals* of each action, i.e., the time interval in which performing the action will cause the agent to collide with another agent. The constraints in CCBS are represented by pairs of actions and unsafe intervals, meaning that an action cannot be executed in its unsafe interval. For the low-level solver, CCBS uses a version of SIPP (Phillips and Likhachev 2011) adapted to handle CCBS constraints.

Our $T$R-CCBS differs from CCBS in how it identifies and resolves conflicts. $T$R-CCBS identifies $T$-delay conflicts. To resolve $T$-delay conflicts, a simple idea is to add $T$ to the unsafe interval found by CCBS. However, if two agents do not collide when they are not delayed, there will not be any unsafe interval by CCBS. Therefore, a new collision detection mechanism is required to handle $T$-delay conflicts.

**Identifying $T$-delay conflicts**  For each node $N$ in the CT, $T$R-CCBS scans its joint plan $N.\pi$ for $T$-delay conflicts, which is further described in the next section.

**Resolving $T$-delay conflicts**  Similar to CBS, $T$R-CCBS runs a best-first search of the CT, expanding in every iteration the CT node $N$ that has the joint plan with the lowest cost (if $N$ is a non-goal node). $T$R-CCBS chooses a $T$-delay conflict in $N.\pi$ and generates successor nodes of $N$ to resolve it. To resolve a $T$-delay conflict $\langle a_i, t_i, a_j, t_j \rangle$, $T$R-CCBS computes the *unsafe interval* of each action the other action. The unsafe interval of action $a_i$ with respect to $a_j$ is the maximal time interval $[t_i, t_i^u)$ such that for each $t \in [t_i, t_i^u)$, there exists a value $\Delta \in [0, T]$ so that executing $a_i$ at time $t$ will lead to a collision with executing $a_j$ at time $t_j + \Delta$. The unsafe interval of action $a_j$ with respect to $a_i$ is the maximal time interval $[t_j, t_j^u)$ such that for each $t \in [t_j, t_j^u)$, there exists a value $\Delta \in [0, T]$ so that executing $a_j$ at time $t + \Delta$ will cause a collision with executing $a_i$ at time $t_i$. $T$R-CCBS generates two successor nodes $N_i$ and $N_j$ for $N$, and adds to $N_i$ (resp. $N_j$) a new constraint that agent $i$ (resp. $j$) cannot execute $a_i$ (resp. $a_j$) in its unsafe interval $[t_i, t_i^u)$ (resp. $[t_j, t_j^u)$). Then, the low-level solver runs the adapted SIPP to find new plans for agents $i$ and $j$ in $N_i$ and $N_j$ respectively that satisfy the additional constraints.

$T$R-CCBS uses a similar method to CCBS for computing unsafe intervals based on collision detection results: it iteratively carries out collision detection by shifting the start time of $a_i$ or $a_j$ until no collision is found (Andreychuk et al. 2022).

**Theoretical properties**  $T$R-CCBS is sound, complete, and optimal. Soundness follows from performing conflict detection for the joint plan of every CT node and Lemma 1. Our proof of completeness and optimality is based on

the following lemma from Andreychuk et al. (2022) and the sound pair of constraints defined by Atzmon et al. (2018).

**Lemma 2.**  Running the adapted SIPP from (Andreychuk et al. 2022) with a set of CCBS (or $T$R-CCBS) constraints returns the lowest-cost path that satisfies these constraints.

**Definition 4** (Sound Pair of Constraints).  A pair of constraints is sound if every $T$-robust plan satisfies at least one of these constraints.

**Lemma 3.**  For a $T$-delay conflict $\langle a_i, t_i, a_j, t_j \rangle$ and corresponding unsafe intervals $[t_i, t_i^u)$ and $[t_j, t_j^u)$, the pair of constraints forbidding agents $i$ and $j$ to execute actions $a_i$ and $a_j$ in $[t_i, t_i^u)$ and $[t_j, t_j^u)$ respectively is a sound pair of constraints.

*Proof.*  Assume on the contrary that there exists a $T$-robust plan in which agent $i$ executes $a_i$ at time $t_i + \delta_i$ for some $\delta_i \in [0, t_i^u - t_i)$, and agent $j$ executes $a_j$ at time $t_j + \delta_j$ for some $\delta_j \in [0, t_j^u - t_j)$. By Lemma 1, $\langle a_i, t_i + \delta_i, a_j, t_j + \delta_j \rangle$ is not a $T$-delay conflict.

By the definition of $t_j^u$, for each $t \in [t_j, t_j^u)$, there exists a value $\Delta \in [0, T]$ such that executing $a_j$ at time $t + \Delta$ will cause a collision with executing $a_i$ at time $t_i$, i.e., $\langle a_i, t_i, a_j, t \rangle$ is a $T$-delay conflict by Definition 3. This implies that for each $\delta \in [0, \delta_j]$ and each $t \in [t_j + \delta, t_j^u + \delta)$, $\langle a_i, t_i + \delta, a_j, t \rangle$ is also a $T$-delay conflict. It follows from $t_j + \delta_j \in [t_j + \delta, t_j^u + \delta)$ that $\langle a_i, t_i + \delta, a_j, t_j + \delta_j \rangle$ is a $T$-delay conflict. Since $\langle a_i, t_i + \delta_i, a_j, t_j + \delta_j \rangle$ is not a $T$-delay conflict, we must have $\delta_i \notin [0, \delta_j]$, which indicates $\delta_i > \delta_j$.

By the definition of $t_i^u$, for each $t \in [t_i, t_i^u)$, there exists a value $\Delta \in [0, T]$ such that executing $a_i$ at time $t$ will lead to a collision with executing $a_j$ at time $t_j + \Delta$, i.e., $\langle a_i, t, a_j, t_j \rangle$ is a $T$-delay conflict by Definition 3. This implies that for each $\delta \in [0, \delta_i]$ and for each $t \in [t_i + \delta, t_i^u + \delta)$, $\langle a_i, t, a_j, t_j + \delta \rangle$ is also a $T$-delay conflict. It follows from $t_i + \delta_i \in [t_i + \delta, t_i^u + \delta)$ that $\langle a_i, t_i + \delta_i, a_j, t_j + \delta \rangle$ is a $T$-delay conflict. Since $\langle a_i, t_i + \delta_i, a_j, t_j + \delta_j \rangle$ is not a $T$-delay conflict, we must have $\delta_j \notin [0, \delta_i]$, which indicates $\delta_j > \delta_i$.

As a result, a contradiction arises.  $\square$

**Theorem 1.**  $T$R-CCBS is guaranteed to return an optimal $T$-robust plan if there exists one.

*Proof.*  The proof, based on Lemma 2 and Lemma 3, is similar to Andreychuk et al. (2022)'s proof for CCBS. Consider any CT node $N$. Let $N_1$ and $N_2$ be the successors of $N$, generated by a sound pair of constraints $C_1$ and $C_2$ respectively (Lemma 3). Let $\pi(N)$ denote all $T$-robust plans that satisfy $N.constraints$. By the definition of a sound pair of constraints (Definition 4), it holds that $\pi(N) = \pi(N_1) \cup \pi(N_2)$. Thus, splitting a CT node does not cause it to lose any $T$-robust plan. Due to (1) Lemma 2, (2) the best-first search of the CT in terms of the cost, and (3) the fact that adding constraints can never reduce the cost, $T$R-CCBS is guaranteed to return an optimal $T$-robust plan if there exists one.  $\square$

## Collision Detection under $T$-Delay

The goal of collision detection is to find out whether the geometry shapes of two agents will overlap when performing

their actions. Recall that to solve $T$-robust MAPF, actions are timed. Our general approach is to first derive the time interval in which the shapes of two moving agents overlap (we call it the *collision interval*) and then check if the collision interval intersects the time intervals of the actions to infer whether collisions will occur during the actions.

## Sampling Approach

To check for $T$-delay conflicts, we first present a discretized sampling approach that proceeds by sampling for possible delays in each agent and checking if the agents collide. Consider two timed actions $(a_i, t_i)$ and $(a_j, t_j)$ of two circular agents $i$ and $j$ respectively. Let $\mathbf{T}$ be the set of discretized delay samples in $[0, T]$. For each $\Delta_i \in \mathbf{T}$ and each $\Delta_j \in \mathbf{T}$, we derive the collision interval $[\tau^-, \tau^+]$ in which the geometry shapes of agents $i$ and $j$ overlap, if they start performing actions $a_i$ and $a_j$ at times $t_i + \Delta_i$ and $t_j + \Delta_j$ respectively. The detailed derivation will be given later in the section titled "Collision detection between two moving circles". Assume that the durations of actions $a_i$ and $a_j$ are $d_i$ and $d_j$ respectively. Under delays $\Delta_i$ and $\Delta_j$, the time intervals of the actions are $[t_i + \Delta_i, t_i + d_i + \Delta_i)$ and $[t_j + \Delta_j, t_j + d_j + \Delta_j)$. Hence, a collision would really occur if and only if these two intervals and the collision interval found intersect:

$$[t_i + \Delta_i, t_i + \Delta_i + d_i) \cap [t_j + \Delta_j, t_j + \Delta_j + d_j) \cap [\tau^-, \tau^+) \neq \emptyset.$$

A drawback of the sampling approach is that it has an increasing computational complexity with respect to the number of delay samples and hence $T$ assuming a constant sampling size per unit delay. Next, we propose to solve for the *exact* collision interval between two agents that may be delayed. The key benefit of this method is that the computational complexity of the collision detection is constant to $T$.

## Extruding Time Delay Dimension

We propose to use Constructive Solid Geometry (CSG) for collision detection under $T$-delay. CSG forms objects from primitives by combining them with set-theoretic operations (Requicha and Rossignac 1992). By treating the time domain as an additional polygonal dimension, the polygons can be extruded into the time dimension, after which a static polygonal intersection check is applied (Walker and Sturtevant 2019). To check for $T$-delay conflicts, we extrude the agent shapes into the time-delay dimension and perform dynamic collision detection between extruded geometries.

Assume that the agents have circular shapes and move with constant velocities, and each action is a linear motion from the start location to the target location. We can extrude the circular agents into a *stadium*, which is a two-dimensional geometric shape composed of a rectangle with semicircles at a pair of opposite sides. The stadium represents all possible areas occupied by the agent under the delay. In general, assuming the center point of the agent without delay is $p$, the maximum delayed center point of the agent is $p^d = p - vT$, where $v$ is the velocity of the agent. Figure 1 shows a stadium formed from two circles centered at $p$ and $p^d$.[1] We can then perform collision detection be-

---

[1]If it is a wait action, then $v = 0$, $p$ and $p^d$ are the same, so that the stadium degenerates to the circular shape of the agent.

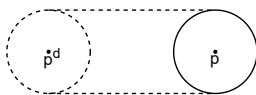

Figure 1: A stadium formed from a circular agent ($p$) and its delayed position ($p^d$).

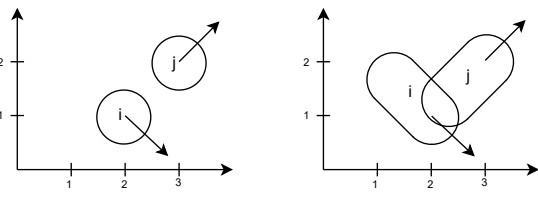

(a) Agents are not delayed     (b) Agents may be delayed.

Figure 2: Collision detection between two circular agents.

tween the stadium geometry formed by each agent. Figure 2 illustrates the collision detection for (a) agents that are not delayed and (b) agents that can be delayed (by extruding the circular geometry into a stadium geometry).

Special considerations, however, need to be taken at the beginning and end of an action, because both $p$ and $p^d$ cannot go beyond the start and target locations of the action. Consider a timed action $(a_i, t_i)$ of agent $i$. Let $d_i$ be the duration of the action, $v$ be the velocity of the agent, and $p_0$ be the start center point of the agent (then the target center point of the agent is $p_0 + vd_i$). Figure 3 illustrates the evolution of the stadium for the action.

(1) Initially, both center points $p$ and $p^d$ are at the start location $p_0$, so the stadium degenerates to a circle.

(2) During the initial $T$ duration of the action, the length of the rectangle in the stadium gradually grows from 0 to $vT$, i.e., for each $t \in [t_i, t_i + \min\{T, d_i\})$, the center point without delay will be $p = p_0 + v(t - t_i)$ and the maximum delayed center point will remain at $p^d = p_0$.

(3) There are two cases for the time interval $[t_i + \min\{T, d_i\}, t_i + \max\{T, d_i\})$:

    $d_i \geq T$: In this case, $[t_i + \min\{T, d_i\}, t_i + \max\{T, d_i\}) = [t_i + T, t_i + d_i)$. During this time interval, the stadium stops growing (with the rectangle reaching the maximum length of $vT$) and continues to move towards the target location, i.e., for each $t \in [t_i + T, t_i + d_i)$, the center point without delay will be $p = p_0 + v(t - t_i)$ and the maximum delayed center point will be $p^d = p - vT = p_0 + v(t - t_i - T)$.

    $d_i < T$: In this case, $[t_i + \min\{T, d_i\}, t_i + \max\{T, d_i\}) = [t_i + d_i, t_i + T)$. During this time interval, the stadium stops growing (with the rectangle reaching the maximum length of $vd_i$) because the agent has reached the target location, i.e., for each $t \in [t_i + d_i, t_i + T)$, the center point without delay will remain at $p = p_0 + vd_i$ and the maximum delayed center point will remain at $p^d = p_0$.

(4) Beyond the action completion without delay, the length of the rectangle in the stadium gradually shrinks from $vT$

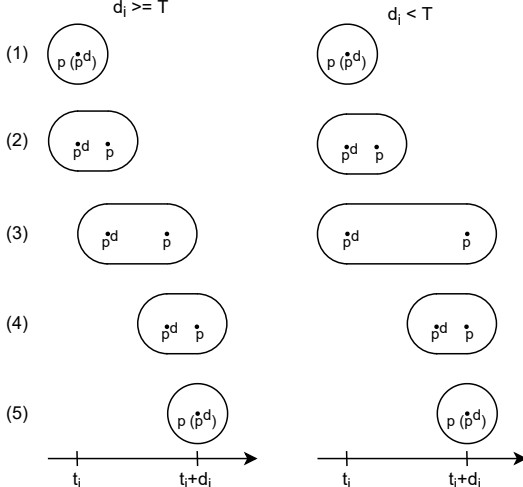

Figure 3: Evolution of the stadium over time for an action.

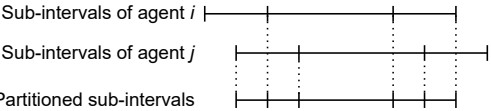

Figure 4: Sub-intervals from agents $i$ and $j$ are further divided into smaller sub-intervals.

to 0, i.e., for each $t \in [t_i + \max\{T, d_i\}, t_i + d_i + T)$, the center point without delay will remain at $p = p_0 + vd_i$ and the maximum delayed center point will be $p^d = p - v(t_i + d_i + T - t) = p_0 - v(t_i + T - t)$.

(5) Finally, both center points $p$ and $p^d$ will be at the target location $p_0 + vd_i$, so the stadium again degenerates to a circle.

Conceptually, a stadium is constructed from a set union of two circles and a rectangle. Let us represent the stadium geometry of agent $i$ as $G_i = \{c_i, r_i, c_i^d\}$, consisting of a circle $c_i$ (centered at $p$), a rectangle $r_i$, and another circle $c_i^d$ (centered at $p^d$). Based on the above evolution, we can partition the time interval $[t_i, t_i + d_i + T)$ into three sub-intervals $[t_i, t_i + \min\{T, d_i\})$, $[t_i + \min\{T, d_i\}, t_i + \max\{T, d_i\})$ and $[t_i + \max\{T, d_i\}, t_i + d_i + T)$. In each sub-interval, the circles $c_i$ and $c_i^d$ move with constant velocities. The rectangle $r_i$ also moves with constant velocity, in the sense that all four vertices move along the same direction with constant velocities. In the second sub-interval, since the stadium has grown to its maximum length, all four vertices move at the same velocity. In the first and last sub-intervals, due to the growing or shrinking of the stadium, a pair of vertices move at a different velocity from the other pair of vertices at the opposite side.

To check for $T$-delay conflicts between two timed actions $(a_i, t_i)$ and $(a_j, t_j)$ of agents $i$ and $j$ with durations $d_i$ and $d_j$, we perform collision detection between their stadium geometries $G_i = \{c_i, r_i, c_i^d\}$ and $G_j = \{c_j, r_j, c_j^d\}$, focusing on the intersection between the time intervals $[t_i, t_i + d_i + T)$ and $[t_j, t_j + d_j + T)$. Figure 4 shows the combination of partitioning of $[t_i, t_i + d_i + T)$ and $[t_j, t_j + d_j + T)$, which divides the intersection $[t_i, t_i + d_i + T) \cap [t_j, t_j + d_j + T)$ into smaller sub-intervals in which all the geometric primitives in $G_i$ and $G_j$ move at constant velocities. In each sub-interval, we can break down the collision detection into that between moving geometric primitives, i.e., among moving circles and rectangles. A collision only occurs if any collision interval found

from the primitives intersects the sub-interval. In what follows, we describe the collision detection between (1) two circles, (2) a circle and a rectangle, and (3) two rectangles.

**(1) Collision detection between two moving circles** The collision detection between two moving circles is based on Guy and Karamouzas (2019)'s descriptions. Given two circular agents $i$ and $j$, the distance between their centers at time $t$ is given by:

$$dist(t) = ||(p_i + v_i t) - (p_j + v_j t)||,$$

where $p_i$ and $p_j$ are the center points of the circles at time 0, and $v_i$ and $v_j$ are the velocities of the agents. Squaring and expanding leads to the following quadratic equation for $t$:

$$dist^2(t) = (v_\Delta \cdot v_\Delta)t^2 + 2(p_\Delta \cdot v_\Delta)t + (p_\Delta \cdot p_\Delta), \quad (1)$$

where $p_\Delta = p_j - p_i$, $v_\Delta = v_j - v_i$ and $\cdot$ is the dot product of vectors. Via substitution, equation (1) can be rewritten as:

$$dist^2(t) = at^2 + bt + c_0,$$

where $a = v_\Delta \cdot v_\Delta$, $b = 2(p_\Delta \cdot v_\Delta)$ and $c_0 = p_\Delta \cdot p_\Delta$.

A collision occurs when the squared distance between the agents is less than or equal to the squared sum of their radii, $r$:

$$dist^2(t) \leq (2r)^2.$$

Thus, we have the following equation for collision detection:

$$at^2 + bt + c \leq 0,$$

where $c = p_\Delta \cdot p_\Delta - (2r)^2$. We can solve it using the quadratic formula to determine the start and end times of the collision interval between two agents:

$$\left[ \frac{-b - \sqrt{b^2 - 4ac}}{2a}, \frac{-b + \sqrt{b^2 - 4ac}}{2a} \right).$$

The above formulation assumes that the two agents start their actions at the same time while at positions $p_i$ and $p_j$. If they start their actions at different times $t_i$ and $t_j$, we need to align their positions to the same time instant before performing the collision detection. Specifically, if $t_i < t_j$, the center point of agent $i$ is first moved to $p_i' = p_i + v_i(t_j - t_i)$ and collision detection is then performed with $p_i'$ and $p_j$.

**(2) Collision detection between a moving circle and a moving rectangle** First, we study the collision detection between a circle and an axis-aligned rectangle. Next, we extend it to consider a moving circle and a moving rectangle.

For an axis-aligned rectangle, we first find the point on the rectangle edges that is the closest to the circle's center and then check whether the point is located in the circle. The former can be done by clamping the coordinates of the circle's

center $p_i$ to the coordinates of the rectangle's vertices (let $p_j$ be the lower-left vertex of the rectangle, $w$ and $l$ be its width and length):

$$p_\Delta[x] = p_i[x] - \max(p_j[x], \min(p_i[x], p_j[x] + w)),$$
$$p_\Delta[y] = p_i[y] - \max(p_j[y], \min(p_i[y], p_j[y] + l)).$$

A collision occurs if $p_\Delta \cdot p_\Delta \leq r^2$, where $r$ is the radius of the circle.

To extend this formulation to derive the collision interval between the moving circle and rectangle, we form a set of positions by combining the elements in $p_\Delta[x]$ and $p_\Delta[y]$:

$$P_x = \{p_j[x], p_i[x], p_j[x] + w\},$$
$$P_y = \{p_j[y], p_i[y], p_j[y] + l\},$$
$$P' = \{p' | p' = \langle p'[x], p'[y] \rangle, p'[x] \in P_x, p'[y] \in P_y\}.$$

For each position $p' \in P'$, we find its the corresponding velocity $v'$. Then, we substitute $p_\Delta = p_i - p'$ and $v_\Delta = v_i - v'$ into equation (1) and solve the equation $dist^2(\tau) \leq r^2$ to get a collision interval. Then, we combine these intervals by taking the minimum start time and the maximum end time to obtain the complete collision interval between a moving circle and a moving axis-aligned rectangle.

If the rectangle is not axis-aligned, we can rotate the rectangle vertices to align with the axis, and rotate the rectangle and circle velocities accordingly, before applying the above formulation to derive the collision interval.

**(3) Collision detection between two moving rectangles**
We can utilize the Separating Axis Theorem (SAT) to perform collision detection between two moving rectangles (Boyd and Vandenberghe 2004). The velocities of the rectangles are used to determine the separating axes as the moving directions of the rectangles decide where two rectangles will collide first. The separating axes, $A$, are a set consisting of the normalized vectors of the velocities, $||v_i||$ and $||v_j||$, their orthogonals, $\perp ||v_i||$ and $\perp ||v_j||$, and the sum of the velocities, $||v_i + v_j||$ and $\perp ||v_i + v_j||$. The sum of the velocities is required for the case where $v_i$ is orthogonal to $v_j$.

$$A = \begin{cases} \{||v_i + v_j||, \perp ||v_i + v_j||\} & \text{if } v_i \perp v_j, \\ \{||v_i||, \perp ||v_i||, ||v_j||, \perp ||v_j||\} & \text{otherwise.} \end{cases}$$

Let $P_i$ and $P_j$ be the sets of vertices of two rectangles. For each pair of vertices $p_i \in P_i$ and $p_j \in P_j$, we find their corresponding velocities $v_i$ and $v_j$. Then, we project the vertices and velocities onto each separating axis, $a \in A$:

$$p'_i = p_i \cdot a, \qquad v'_i = v_i \cdot a,$$
$$p'_j = p_j \cdot a, \qquad v'_j = v_j \cdot a.$$

After that, for each pair of projected vertices $p'_i$ and $p'_j$, we find the time instants when the moving points overlap (i.e., the distance between the points is equal to zero) to derive a collision interval. Specifically, we substitute $p_\Delta = p'_i - p'_j$ and $v_\Delta = v'_i - v'_j$ into equation (1) and solve the equation $dist^2(\tau) = 0$ to get a collision interval. Then, we combine these intervals by taking the minimum start time and the maximum end time to obtain the complete collision interval between two moving rectangles.

## Experiments

We implemented $T$R-CCBS on top of CCBS (https://github.com/pathplanning/continuous-cbs) and conducted experiments using the same experimental settings of Andreychuk et al. (2022). Agents move in $2^k$-neighborhood grids. In a move action, agents can move from the center of one grid cell to the center of any grid cell located in their $2^k$ neighborhood, where $k$ is a parameter (Rivera, Hernández, and Baier 2017). The size of every cell is $1 \times 1$, and the shape of every agent is a circle of radius $\sqrt{2}/4$. Moves are allowed only if the agent can move safely to the target cell without colliding with other agents or obstacles, considering the geometry shapes of the agents and obstacles. The moving speeds of agents are 1 per unit time, so the duration of a move action corresponds to the Euclidean distance between the centers of the start and target cells. We used the *makespan* as the objective function in pathfinding. Constructive Solid Geometry (CSG) was used for collision detection, unless stated otherwise. The experiments were run on an Intel i9-12900K 5.6GHz processor with 32GB memory.

### Open Grids

In the first set of experiments, we used a $10 \times 10$ open grid. For every number of agents, we created 250 different problems by placing agents' start and goal locations randomly.

**Increasing number of agents** We performed experiments with $2 \sim 24$ agents. Each problem was solved with $T$R-CCBS for $k = 2, 3, 4, 5$, and $T$-delay $= 0, 0.5, 1, 1.5, 2$. Note that $T = 0$ is the standard CCBS.

The top half of Table 1 presents the success rate, i.e., the ratio of problems solved by $T$R-CCBS within a time limit of 60 seconds. Every row shows the results for a distinct number of agents, as indicated in the left-most column. Every five columns show the results for a $2^k$-neighborhood ranging from $k = 2 \sim 5$. Each of the five columns shows the success rate for a distinct $T$-delay. Data points marked by "-" indicate settings where the success rate is 0. The bottom half of Table 1 presents the cost (makespan) of the output plan for one particular problem (of 250 problems tested), where data points marked by "-" indicate that no solution was found within the time limit.

It can be seen from Table 1 that increasing the number of agents leads to a lower success rate, which was also shown for the original CCBS (Andreychuk et al. 2022). A larger number of agents result in a denser environment, which is more difficult to solve as more potential conflicts need to be resolved. Increasing the number of agents under a higher $T$-delay exhibits a sharper decrease in the success rate. This is especially prominent for $T = 2$, where no solution was found within the time limit for more than $12 \sim 17$ agents. Due to the higher delay, agents need to reserve more space to avoid conflicts, which increases the hardness of pathfinding.

Table 1 also shows that increasing $k$ yields a lower success rate. For example, for $T = 0$ and 24 agents, the success rate for $k = 2$ is 0.688, while that for $k = 5$ is only 0.12. A larger $k$ increases the branching factor (more paths available), making the optimal plan harder to find.

| Success Ratio | | | | | | | | | | | | | | | | | | | | |
|---|---|---|---|---|---|---|---|---|---|---|---|---|---|---|---|---|---|---|---|---|
| | $k=2$ | | | | | $k=3$ | | | | | $k=4$ | | | | | $k=5$ | | | | |
| Agents \ $t_d$ | 0.0 | 0.5 | 1.0 | 1.5 | 2.0 | 0.0 | 0.5 | 1.0 | 1.5 | 2.0 | 0.0 | 0.5 | 1.0 | 1.5 | 2.0 | 0.0 | 0.5 | 1.0 | 1.5 | 2.0 |
| 2 | 1.000 | 0.996 | 0.996 | 0.992 | 0.968 | 1.000 | 1.000 | 0.992 | 0.976 | 0.972 | 1.000 | 0.992 | 0.988 | 0.980 | 0.976 | 1.000 | 0.996 | 0.980 | 0.976 | 0.968 |
| 3 | 1.000 | 0.992 | 0.988 | 0.972 | 0.940 | 1.000 | 0.996 | 0.976 | 0.940 | 0.912 | 1.000 | 0.980 | 0.964 | 0.928 | 0.924 | 1.000 | 0.984 | 0.948 | 0.940 | 0.916 |
| 4 | 1.000 | 0.984 | 0.968 | 0.932 | 0.904 | 1.000 | 0.996 | 0.944 | 0.908 | 0.836 | 1.000 | 0.964 | 0.904 | 0.868 | 0.848 | 1.000 | 0.964 | 0.896 | 0.880 | 0.860 |
| 5 | 1.000 | 0.928 | 0.916 | 0.852 | 0.772 | 1.000 | 0.952 | 0.856 | 0.768 | 0.692 | 1.000 | 0.896 | 0.808 | 0.760 | 0.728 | 1.000 | 0.908 | 0.812 | 0.796 | 0.748 |
| 6 | 1.000 | 0.908 | 0.892 | 0.804 | 0.704 | 1.000 | 0.944 | 0.796 | 0.684 | 0.608 | 1.000 | 0.868 | 0.740 | 0.664 | 0.584 | 1.000 | 0.880 | 0.752 | 0.708 | 0.604 |
| 7 | 1.000 | 0.872 | 0.820 | 0.712 | 0.616 | 1.000 | 0.896 | 0.696 | 0.576 | 0.460 | 1.000 | 0.816 | 0.636 | 0.524 | 0.456 | 1.000 | 0.824 | 0.656 | 0.552 | 0.456 |
| 8 | 1.000 | 0.832 | 0.776 | 0.612 | 0.508 | 1.000 | 0.856 | 0.556 | 0.388 | 0.324 | 1.000 | 0.756 | 0.492 | 0.356 | 0.312 | 1.000 | 0.732 | 0.484 | 0.360 | 0.284 |
| 9 | 1.000 | 0.772 | 0.692 | 0.472 | 0.388 | 1.000 | 0.780 | 0.468 | 0.300 | 0.228 | 1.000 | 0.664 | 0.388 | 0.252 | 0.188 | 0.996 | 0.624 | 0.352 | 0.228 | 0.152 |
| 10 | 1.000 | 0.700 | 0.624 | 0.356 | 0.300 | 1.000 | 0.704 | 0.360 | 0.204 | 0.124 | 1.000 | 0.580 | 0.284 | 0.152 | 0.096 | 0.996 | 0.532 | 0.228 | 0.128 | 0.072 |
| 11 | 1.000 | 0.636 | 0.536 | 0.260 | 0.216 | 1.000 | 0.640 | 0.252 | 0.116 | 0.056 | 0.996 | 0.472 | 0.192 | 0.096 | 0.056 | 0.980 | 0.420 | 0.164 | 0.088 | 0.032 |
| 12 | 0.996 | 0.540 | 0.436 | 0.172 | 0.132 | 0.996 | 0.536 | 0.176 | 0.060 | 0.028 | 0.992 | 0.392 | 0.128 | 0.040 | 0.028 | 0.956 | 0.268 | 0.092 | 0.040 | 0.004 |
| 13 | 0.972 | 0.456 | 0.356 | 0.140 | 0.088 | 0.992 | 0.440 | 0.104 | 0.024 | 0.004 | 0.988 | 0.292 | 0.060 | 0.012 | - | 0.956 | 0.208 | 0.044 | 0.008 | - |
| 14 | 0.956 | 0.404 | 0.300 | 0.096 | 0.068 | 0.996 | 0.360 | 0.072 | 0.016 | 0.004 | 0.980 | 0.208 | 0.036 | 0.004 | - | 0.884 | 0.132 | 0.024 | 0.004 | - |
| 15 | 0.956 | 0.340 | 0.260 | 0.060 | 0.040 | 0.992 | 0.264 | 0.044 | 0.004 | - | 0.968 | 0.140 | 0.016 | 0.004 | - | 0.836 | 0.080 | 0.016 | 0.004 | - |
| 16 | 0.936 | 0.236 | 0.176 | 0.016 | 0.004 | 0.988 | 0.180 | 0.020 | 0.004 | - | 0.932 | 0.096 | 0.012 | 0.004 | - | 0.764 | 0.044 | 0.008 | 0.004 | - |
| 17 | 0.928 | 0.188 | 0.128 | 0.012 | 0.004 | 0.964 | 0.124 | 0.004 | - | - | 0.904 | 0.048 | 0.004 | - | - | 0.676 | 0.024 | 0.004 | - | - |
| 18 | 0.896 | 0.156 | 0.096 | 0.004 | - | 0.948 | 0.080 | 0.004 | - | - | 0.840 | 0.036 | 0.004 | - | - | 0.568 | 0.012 | - | - | - |
| 19 | 0.872 | 0.120 | 0.068 | 0.004 | - | 0.936 | 0.044 | - | - | - | 0.760 | 0.016 | - | - | - | 0.500 | - | - | - | - |
| 20 | 0.852 | 0.068 | 0.044 | 0.004 | - | 0.916 | 0.024 | - | - | - | 0.676 | - | - | - | - | 0.424 | - | - | - | - |
| 21 | 0.800 | 0.040 | 0.028 | - | - | 0.848 | 0.012 | - | - | - | 0.576 | - | - | - | - | 0.352 | - | - | - | - |
| 22 | 0.768 | 0.028 | 0.020 | - | - | 0.816 | 0.004 | - | - | - | 0.496 | - | - | - | - | 0.284 | - | - | - | - |
| 23 | 0.720 | 0.016 | 0.012 | - | - | 0.736 | - | - | - | - | 0.416 | - | - | - | - | 0.192 | - | - | - | - |
| 24 | 0.688 | 0.012 | 0.004 | - | - | 0.672 | - | - | - | - | 0.344 | - | - | - | - | 0.120 | - | - | - | - |
| Plan Cost for a Particular Problem | | | | | | | | | | | | | | | | | | | | |
| 2 | 6 | 6 | 6 | 6 | 6 | 4.24 | 4.24 | 4.24 | 4.24 | 4.24 | 4.24 | 4.24 | 4.24 | 4.24 | 4.24 | 4.24 | 4.24 | 4.24 | 4.24 | 4.24 |
| 3 | 15 | 15 | 15 | 15 | 15 | 12.66 | 12.66 | 12.66 | 12.66 | - | 11.94 | 11.94 | - | - | - | 11.72 | 11.72 | - | - | - |
| 4 | 15 | 15 | 15 | 15 | 15 | 12.66 | 12.66 | 12.66 | 12.66 | - | 11.94 | 11.94 | - | - | - | 11.72 | 11.72 | - | - | - |
| 5 | 15 | 15 | 15 | 17 | - | 14.01 | 14.51 | 14.66 | 14.66 | - | 13.65 | 14.15 | - | - | - | 13.50 | 13.99 | - | - | - |
| 6 | 15 | 15 | 17 | - | - | 14.01 | 14.59 | 14.66 | 14.66 | - | 13.65 | 14.21 | - | - | - | 13.50 | 14.06 | - | - | - |
| 7 | 15 | 15 | 17 | - | - | 14.01 | 14.59 | 14.66 | 14.66 | - | 13.65 | 14.21 | - | - | - | 13.50 | 14.06 | - | - | - |
| 8 | 16 | 16.50 | 17 | - | - | 14.01 | 14.59 | - | - | - | 13.65 | 14.21 | - | - | - | 13.50 | - | - | - | - |
| 9 | 16 | 16.50 | 17 | - | - | 14.01 | 14.59 | - | - | - | 13.65 | 14.21 | - | - | - | 13.50 | - | - | - | - |
| 10 | 16 | 16.50 | 17 | - | - | 14.01 | 14.59 | - | - | - | 13.65 | 14.21 | - | - | - | 13.50 | - | - | - | - |
| 11 | 16 | 16.50 | 17 | - | - | 14.01 | 14.59 | - | - | - | 13.65 | 14.21 | - | - | - | 13.50 | - | - | - | - |
| 12 | 16 | 16.50 | 17 | - | - | 14.01 | 14.59 | - | - | - | 13.65 | 14.21 | - | - | - | 13.50 | - | - | - | - |
| 13 | 16 | 16.50 | 17 | - | - | 14.01 | - | - | - | - | 13.65 | - | - | - | - | 13.50 | - | - | - | - |
| 14 | 20 | 20 | 20 | - | - | 14.14 | - | - | - | - | 14.14 | - | - | - | - | 14.14 | - | - | - | - |
| 15 | 20 | 20 | 20 | - | - | 14.14 | - | - | - | - | 14.14 | - | - | - | - | 14.14 | - | - | - | - |
| 16 | 20 | - | - | - | - | 14.14 | - | - | - | - | 14.14 | - | - | - | - | - | - | - | - | - |
| 17 | 20 | - | - | - | - | 14.33 | - | - | - | - | 14.33 | - | - | - | - | - | - | - | - | - |
| 18 | 20 | - | - | - | - | 14.33 | - | - | - | - | 14.33 | - | - | - | - | - | - | - | - | - |
| 19 | 20 | - | - | - | - | 14.33 | - | - | - | - | 14.33 | - | - | - | - | - | - | - | - | - |
| 20 | 20 | - | - | - | - | 14.33 | - | - | - | - | - | - | - | - | - | - | - | - | - | - |
| 21 | 20 | - | - | - | - | 14.33 | - | - | - | - | - | - | - | - | - | - | - | - | - | - |
| 22 | 20 | - | - | - | - | 14.33 | - | - | - | - | - | - | - | - | - | - | - | - | - | - |
| 23 | 20 | - | - | - | - | 14.33 | - | - | - | - | - | - | - | - | - | - | - | - | - | - |
| 24 | 20 | - | - | - | - | 14.33 | - | - | - | - | - | - | - | - | - | - | - | - | - | - |

Table 1: Results for open grid scenario across an increasing number of agents.

As seen from Table 1, the cost of the plan increases with the number of agents due to the higher density of the environment. For example, when $k = 2$, the cost is 6 for 2 agents, and increases to 20 for 14 agents and beyond. Similarly, the cost of the plan also increases with the $T$-delay and hence the hardness of pathfinding. However, increasing $k$ reduces the cost of the plan. For 2 agents, the cost reduces from 6 for $k = 2$, down to 4.24 for $k = 3$ and beyond. This is because increasing $k$ means that there are more direct routes available between the start and goal locations and thus the paths for the agents can be shorter. Overall, increasing $k$ introduces two diverse effects on pathfinding: the resulting search space for the low-level solver becomes shallower but wider.

**Increasing $T$-delay** Next, we evaluated a wider range of the $T$-delay. We only performed the experiments with 5, 10, and 15 agents. Again, we created 250 different problems for every number of agents. Each problem was solved with $TR$-CCBS for $k = 2, 3, 4$, and $T$-delay $= 0, 0.5, 1, \dots, 6.5, 7$.

Table 2 presents the results. Every row shows the results for a distinct $T$-delay, as indicated in the left-most column. In the top half of the table, every three columns show the

| Success Rate | | | | | | | | | |
|---|---|---|---|---|---|---|---|---|---|
| | $k=2$ | | | $k=3$ | | | $k=4$ | | |
| $T$ \ Agents | 5 | 10 | 15 | 5 | 10 | 15 | 5 | 10 | 15 |
| 0.0 | 1.000 | 0.992 | 0.936 | 1.000 | 1.000 | 0.992 | 1.000 | 1.000 | 0.964 |
| 0.5 | 0.928 | 0.700 | 0.340 | 0.848 | 0.484 | 0.116 | 0.820 | 0.328 | 0.036 |
| 1.0 | 0.916 | 0.616 | 0.260 | 0.780 | 0.272 | 0.044 | 0.752 | 0.204 | 0.016 |
| 1.5 | 0.852 | 0.356 | 0.056 | 0.680 | 0.120 | 0.008 | 0.672 | 0.080 | 0.008 |
| 2.0 | 0.772 | 0.296 | 0.040 | 0.644 | 0.084 | - | 0.656 | 0.052 | 0.004 |
| 2.5 | 0.708 | 0.168 | 0.008 | 0.548 | 0.040 | - | 0.620 | 0.036 | 0.004 |
| 3.0 | 0.692 | 0.148 | 0.004 | 0.528 | 0.028 | - | 0.564 | 0.032 | 0.004 |
| 3.5 | 0.600 | 0.040 | - | 0.464 | 0.016 | - | 0.536 | 0.012 | - |
| 4.0 | 0.564 | 0.032 | - | 0.388 | 0.004 | - | 0.476 | 0.004 | - |
| 4.5 | 0.500 | 0.012 | - | 0.340 | 0.004 | - | 0.420 | 0.004 | - |
| 5.0 | 0.472 | 0.012 | - | 0.308 | - | - | 0.396 | 0.004 | - |
| 5.5 | 0.408 | 0.008 | - | 0.260 | - | - | 0.360 | 0.004 | - |
| 6.0 | 0.364 | 0.008 | - | 0.236 | - | - | 0.360 | - | - |
| 6.5 | 0.344 | - | - | 0.224 | - | - | 0.316 | - | - |
| 7.0 | 0.324 | - | - | 0.176 | - | - | 0.304 | - | - |
| Plan Cost for a Particular Problem | | | | | | | | | |
| 0.0 | 8.00 | 13.00 | 15.00 | 7.41 | 10.66 | 11.83 | 7.24 | 10.13 | 11.54 |
| 0.5 | 8.00 | 13.00 | 15.50 | 7.41 | 11.07 | 13.90 | 7.24 | 10.13 | 13.19 |
| 1.0 | 8.00 | 13.00 | 16.00 | 7.41 | 11.66 | 14.31 | 7.24 | 10.36 | 14.78 |
| 1.5 | 8.00 | 13.00 | 17.00 | 7.41 | 11.66 | 15.66 | 7.24 | 10.36 | 14.78 |
| 2.0 | 8.00 | 13.00 | 18.00 | 7.41 | 11.66 | - | 7.24 | 10.36 | 15.42 |
| 2.5 | 8.00 | 13.00 | 18.00 | 7.41 | 12.41 | - | 7.24 | 10.94 | 15.42 |
| 3.0 | 8.00 | 13.00 | 18.00 | 7.41 | 12.41 | - | 7.24 | 10.36 | 15.42 |
| 3.5 | 8.00 | 13.00 | - | 7.41 | 14.49 | - | 7.24 | 10.36 | - |
| 4.0 | 8.00 | 13.00 | - | 7.41 | 15.66 | - | 7.24 | 14.18 | - |
| 4.5 | 9.00 | 13.00 | - | 7.41 | 15.49 | - | 7.24 | 14.94 | - |
| 5.0 | 9.00 | 13.00 | - | 7.41 | - | - | 7.24 | 14.94 | - |
| 5.5 | 9.00 | 13.00 | - | 7.41 | - | - | 7.24 | 14.94 | - |
| 6.0 | 9.00 | 13.00 | - | 7.41 | - | - | 7.24 | - | - |
| 6.5 | 9.00 | - | - | 7.66 | - | - | 7.24 | - | - |
| 7.0 | 9.00 | - | - | 7.66 | - | - | 7.24 | - | - |

Table 2: Results for open grids across increasing $T$-delay.

| Agents \ $T$ | Success Rate | | | | | | | | | | | | | | |
|---|---|---|---|---|---|---|---|---|---|---|---|---|---|---|---|
| | $k = 2$ | | | | | $k = 3$ | | | | | $k = 4$ | | | | |
| | 0.0 | 0.5 | 1.0 | 1.5 | 2.0 | 0.0 | 0.5 | 1.0 | 1.5 | 2.0 | 0.0 | 0.5 | 1.0 | 1.5 | 2.0 |
| 10 | 0.964 | 0.912 | 0.912 | 0.832 | 0.832 | 0.960 | 0.888 | 0.840 | 0.800 | 0.744 | 0.984 | 0.932 | 0.888 | 0.828 | 0.772 |
| 15 | 0.932 | 0.792 | 0.780 | 0.660 | 0.656 | 0.908 | 0.760 | 0.628 | 0.572 | 0.520 | 0.924 | 0.772 | 0.692 | 0.572 | 0.508 |
| 20 | 0.884 | 0.636 | 0.632 | 0.452 | 0.452 | 0.824 | 0.512 | 0.340 | 0.296 | 0.224 | 0.820 | 0.504 | 0.384 | 0.292 | 0.220 |
| 25 | 0.824 | 0.468 | 0.456 | 0.296 | 0.276 | 0.748 | 0.260 | 0.148 | 0.100 | 0.072 | 0.696 | 0.224 | 0.140 | 0.080 | 0.056 |
| | Plan Cost of a Particular Problem | | | | | | | | | | | | | | |
| 10 | 191.00 | 191.00 | 191.00 | 191.00 | 191.00 | 155.40 | 155.40 | 155.40 | 155.40 | 155.40 | 149.34 | 149.34 | 149.34 | 149.34 | 149.34 |
| 15 | 195.00 | 195.00 | 195.00 | 195.00 | 195.00 | 157.87 | 157.87 | 157.87 | 157.87 | 157.87 | 154.49 | 154.49 | 154.49 | 154.49 | 154.49 |
| 20 | 236.00 | 236.00 | 236.00 | 254.00 | 254.00 | 200.85 | 200.85 | 200.85 | 200.85 | 200.85 | 196.75 | 196.75 | 196.75 | 196.75 | 220.20 |
| 25 | 236.00 | 236.00 | 236.00 | 255.00 | 255.00 | 200.85 | 200.85 | 200.85 | 229.74 | 229.74 | 196.75 | 205.33 | 219.76 | 219.76 | 233.69 |

Table 3: Results for `den520d` map.

success rate for a $2^k$-neighborhood ranging from $k = 2$ to $4$. Each of the three columns shows the success rate for a distinct number of agents. The bottom half of the table shows the plan cost for one particular problem.

Similar to the results in Table 1, increasing the $T$-delay reduces the success rate due to the increased hardness of pathfinding. In general, increasing the $T$-delay does not significantly increase the cost of the plan found. When the number of agents is small, the likelihood of them colliding with each other is low as the environment is sparse. Hence, it is possible that even increasing the $T$-delay, the agents may still not interact with each other. Thus, increasing the $T$-delay may not have a significant effect on the cost.

## Dragon Age Maps

Further experiments were also performed on a larger grid, `den520d`, from the Dragon Age: Origin (DAO) game (available in the `movingai` repository (Sturtevant 2012)). The start and goal locations of agents were chosen randomly to create 250 different problems for every number of agents.

Table 3 presents the results, which are tabulated in a similar manner to Table 1. In general, the same overall trends are observed: increasing the number of agents or $T$-delay reduces the success rate and increases the plan cost; increasing $k$ reduces the success rate and the plan cost. However, when the number of agents is small, the optimal $T$-robust plan often has the same cost as the optimal 0-robust plan as the map is very large. For example, for 10 agents, there is no increase in the plan cost with increasing $T$-delay.

## Comparison between CSG and Sampling

Lastly, we compare the runtime performance of collision detection using CSG and the sampling approach. We repeated the Open Grids experiments for $k = 4$ using the sampling approach for collision detection and collected the average time spent in performing collision detection per CT node. This is to ensure fair comparison as differences in the unsafe interval will influence the number of CT nodes explored, affecting the overall runtime of the search. In Figure 5, the collision detection time (CDT) is plotted against the $T$-delay for different numbers of agents $a = 5, 10, 15$ and different sample sizes per unit delay $s = 10, 20, 30$. Some lines end before the maximum delay of 7, because the success rate (within a time limit of 60 seconds) drops to 0.

As seen from Figure 5, the CDT for CSG remains quite similar as the $T$-delay increases. This implies that the mag-

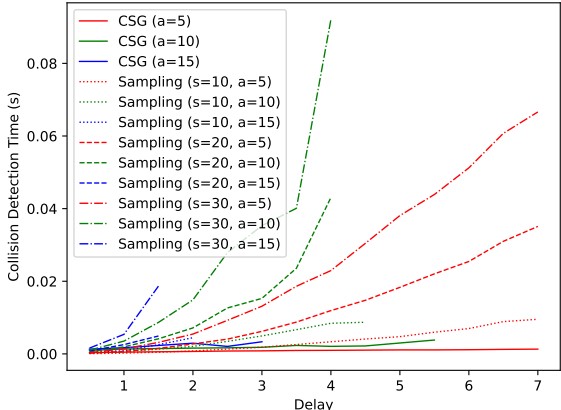

Figure 5: Average collision detection time per CT node against increasing $T$-delay.

nitude of $T$-delay has little impact on the computational complexity of CSG. However, the CDT for the sampling approach increases with the $T$-delay. This is because the number of samples increases proportionally to the $T$-delay. For a sample size of 10, if $T$-delay is below 2, it is possible that sampling has a lower CDT compared to CSG. Other than that, when the $T$-delay is above 1, sampling has a higher CDT compared to CSG. A larger sample size also leads to higher CDT, especially when the $T$-delay is large. The setting of the sample size yields a tradeoff between the computational complexity and the accuracy of collision detection. A larger sample size will increase the computational time of collision detection, whereas a smaller sample size will reduce the accuracy of collision detection.

## Conclusion

In this paper, we have proposed $T$R-CCBS, a sound, complete, and optimal MAPF algorithm that supports continuous time and produces plans that can tolerate a total delay up to $T$ for each agent. $T$R-CCBS extends CCBS by identifying $T$-delay conflicts and resolving them by finding the unsafe intervals for the $T$-delay conflict. We have also presented an exact method for finding the collision interval between agents under the $T$-delay conflict. Our experimental results demonstrate the impact of increasing $T$-delay and agent number on the success rate and plan cost.

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
