# OpenReview forum: "Robust Multi-Agent Pathfinding with Continuous Time"
_icaps-conference.org/ICAPS/2024/Conference — ICAPS 2024_

### Official Review · Reviewer_RQtr · 2024-01-18

**Significance And Importance:** 3
**Soundness:** 4
**Novelty:** 2
**Clarity:** 4
**Overall Evaluation:** 2
**Confidence:** 3

**Weaknesses:**

2: No major or minor weaknesses.

**Contributions Of The Paper:**

The authors propose the T-Robust Continuous-time Conflict-based Search (TT-CCBS) algorithm to solve multi-agent pathfinding (MAPF) problems where agents move in continuous time and the plans of the agents are robust up to a delay of time T.

The authors also propose close-form formulas, adapted from the literature, for collision detection under T-delay.

**Ethical Considerations:**

(1) Not Applicable: The paper does not have any ethical considerations to address

**Nomination For Best Paper:**

No

**Questions For Authors:**

Is there a reason to ever use the sampling approach in detecting collisions? It seems that applications where robustness is important will also require guarantees on the robustness properties, and sampling approaches are unable to provide such guarantees.

**Reproducibility:**

2: Some details are missing, but the paper still appears to be replicable with some effort.

**Strengths Of The Paper:**

The new proposed algorithm fills in a gap in the literature and further improves the applicability of MAPF in the real world, especially for applications where robustness towards delays is important. The new close-form formulas are also notable because they not only compute the possibility of collisions efficiently, but they also do so in an exact manner.

**Weaknesses Of The Paper:**

A *very minor* weakness is that one can view the TR-CCBS algorithm as incremental as it builds on top of (and has a significant overlap with) CCBS. The key differences are in the collision conflict detection and resolution of those conflicts. But the collision detection approaches also strongly adapts findings from the literature.

---

> ### Author Rebuttal · Authors · 2024-01-27
>
> The sampling approach is a general approach that can be applied to any agent shape and kinematic constraints, and it is also simple to compute. It may be desirable to utilize the sampling approach if the closed-form collision detection is too complicated and requires too long to solve. In addition, increasing the sampling frequency will also increase the accuracy of collision detection, and eventually be able to provide the robustness guarantee under certain precision required.

---

### Official Review · Reviewer_vZJH · 2024-01-19

**Significance And Importance:** 1
**Soundness:** 3
**Novelty:** 2
**Clarity:** 4
**Confidence:** 5

**Weaknesses:**

0: Minor weaknesses requiring some work to be addressed for the paper to be accepted.

**Contributions Of The Paper:**

The paper contributes an extension of Continuous Time Conflict Based Search which adds plan robustness up to additive agent delay up to T for each agent. They show that their extension maintains completeness and optimality.

**Ethical Considerations:**

(1) Not Applicable: The paper does not have any ethical considerations to address

**Nomination For Best Paper:**

No

**Overall Evaluation:**

-2: (reject)

**Questions For Authors:**

I'll ask some of the questions I'm pointing out above.

1) For this two dimensional case, would there be additional primitives required beyond a circle and a rectangle to represent a wider assortment of agent shapes?

2) Have you thought about taking the results section in a different direction and using this on a set of physical robots?

3) Are there any applications for this work that you have in mind? It's really neat work, but talking about where you'd see delay at that scale or applications with that many robots would also help to motivate the utility of this great work.

**Reproducibility:**

3: Authors describe the implementation and domains in sufficient detail.

**Strengths Of The Paper:**

The paper is very well written and was enjoyable to read. The authors do a a nice job providing the reader with the background summary of the work that they are extending as well as a few quick mentions of the discrete counter-parts. I think the results are also a good presentation that this method can work for low amounts of expected delay and low agent counts.

**Weaknesses Of The Paper:**

I recognize that there may not be any "apple to apples" work to compare to here, but with results that don't scale particularly well (which is totally to be expected), it's hard to know how really important these results are in this space. It would really strengthen the paper if there was, at the least, a straw man comparison. There is other work, on the discrete side, that used to compare (in the early times) to "resynchronizing" the plan when a delay occurs (which implies a level of centralization admittedly). Or, another straw man could be planning without the delay robustness (CCBS) and then doing replanning when a collision "is about to happen". I do not expect that any of these suggested approaches will work well, but they would provide a helpful context to what you've accomplished.

I really like what you've done with CSG. In your contributions, you suggest you've provided a closed form solution for this problem, but I don't see how this would extend with your derivation beyond simple circular agents? Naturally, you could use this method to identify possible conflicts and then refine that possibility, but that doesn't seem to be addressed in the paper. For this two dimensional case, would there be additional primitives required beyond a circle and a rectangle? Triangles and semi-circles, off the top of my head?

As I said, I think the paper is really well written and quite clear, but it is missing a few things I think are necessary to tell the story you want.

---

> ### Author Rebuttal · Authors · 2024-01-27
>
> 1) To consider T-delay, the geometry of an agent is extruded in the time-delay dimension, and collision detection is performed on the extruded geometries. The CSG approach provides a general framework for collision detection by decomposing the extruded geometry into primitives, where two of them represent the agent at the start and target locations, and the other arises from the trajectory. If the trajectory is a straight line and the agent has a convex shape, the last primitive will most likely be a rectangle or a parallelogram. If an agent has other shapes, we can replace the first two primitives. Determining the type of primitives to decompose the extruded geometry depends on the collision detection method available for the primitives. For example, if the agent is triangular, the extruded geometry will form a convex polygon. Then we can use Separating Axis Theorem (SAT) directly to detect collision between two convex polygons without decomposing the extruded geometry. In addition, to simplify the collision detection method, complex objects can also create bounding circles around the object and perform collision detection using bounding circles where our closed-form solution can be applied.
>
> 2) It will be part of the future work to deploy the TR-CCBS algorithm on a planner for the physical robots. When the integration has been completed, further experimentation on the physical robots will be possible.
>
> 3) The target application is for robots with expensive, unreliable, or no communication during plan execution. This means that the robots are unable to communicate with either the centralized planner or with each other for replanning. Hence, a robust plan is necessary to tolerate possible delays in the plan execution. This work is partly motivated by our discussion with a government agency that is looking into deploying a small number of robots to keep watch over an area (such as a national day event or a subway station) by moving around it.

---

### Official Review · Reviewer_K2fB · 2024-01-22

**Significance And Importance:** 2
**Soundness:** 3
**Novelty:** 3
**Clarity:** 4
**Overall Evaluation:** 2
**Confidence:** 5

**Weaknesses:**

1: Minor weaknesses that are easily fixable.

**Contributions Of The Paper:**

The paper studies a variant of the Multi-Agent Pathfinding problem (MAPF) where time is continuous and each agent may be delayed a limited amount of time. For this problem, an algorithm called TR-CCBS is proposed, which finds the optimal solution that avoids collisions between agents under the possible delays. The paper proves that the algorithm is sound, complete, and optimal. Also, a collision detection mechanism is suggested. Both the algorithm and the suggested collision detection are tested experimentally.

**Ethical Considerations:**

(1) Not Applicable: The paper does not have any ethical considerations to address

**Nomination For Best Paper:**

No

**Questions For Authors:**

1) According to the paper, "A plan is robust to a delay if the delayed agent can continue to execute its remaining plan after the delay without causing any collision". However, assume we have an agent that has been delayed T time and, according to its plan, the agent waits T time at some future vertex. This agent may be able to reduce the future waiting time while keeping the plan robust. Is that correct? Did you think about such a scenario?
2) While the proof of Lemma 1 is missing, I guess it should be almost identical to the similar proof from Atzmon et al., 2020 (Observation 1). Is that correct?
3) TR-CCBS is complete, optimal, and sound. However, what happens if the algorithm is executed on an instance that doesn't have a solution?
4) I couldn't understand how the cost, presented in the experiments section, was computed. Was it averaged out of instances solved for all k values and all delays? For instance, in Table 1, for k=2, t=2.0, and 5 agents, the success rate was 0.772. Why is there no cost value for that setting?

----- Post Rebuttal ----
Thank you for answering my questions in your feedback.
Please add a clarification on the plan cost in the paper (question 4).

**Reproducibility:**

3: Authors describe the implementation and domains in sufficient detail.

**Strengths Of The Paper:**

- The paper is well-written and well-organized. It was easy to follow and understand the paper.
- The paper brings MAPF closer to real-world applications and studies an important problem.
- Some non-trivial ideas are presented, such as adjusting CCBS to avoid collisions while preserving optimality.
- The experimental section evaluates the proposed algorithm on a few k values, number of agents, and number of maps, presenting sufficient experiments for a conference paper.

**Weaknesses Of The Paper:**

In my opinion, there are two main weaknesses (presented below). However, I liked the paper and, with minor modifications (specifically regarding the first weakness), the paper will be ready for publication.

1) Related work - while some relevant related work is not mentioned in the paper and should be mentioned, none of these directly impacts the proposed methods. Moreover, a few citations are missing in the paper. The related work that is relevant to the paper and the missing citations are:
- In the introduction, a few papers are cited as "efficient algorithms have been developed". However, recent papers are not mentioned there, such as: (1) Li et al. "Pairwise symmetry reasoning for multi-agent path finding search". Artificial Intelligence (2021). (2) Gange et al. "Lazy cbs: implicit conflict-based search using lazy clause generation". ICAPS (2019). (3) Lam et al. "Branch-and-cut-and-price for multi-agent path finding". Computers and Operations Research (2022).
- In the related work section, citations for the increasing cost tree search algorithm and for SIPP are missing.
- In the related work section, the following papers should also be mentioned: (1) Chen et al. "Symmetry Breaking for k-Robust Multi-Agent Path Finding". AAAI (2021). (2) Shahar et al. "Safe multi-agent pathfinding with time uncertainty". JAIR (2021). (3) Wagner and Choset. "Path Planning for Multiple Agents under Uncertainty". ICAPS (2017). (4) Atzmon et al. "Probabilistic robust multi-agent path finding". ICAPS (2020).

2) Experimental study - while I find the experiments sufficient for a conference paper, I think it would have been nice if there were results comparing the proposed method with a similar method for a discretized environment. This may show the benefit of keeping the problem continuous.

Minor comments and suggestions -
- In the related work section, what the "expected makespan" is should be explained.
- In the T-Robust MAPF section, the term "timed action" is used, but the first time an action is mentioned in this section, it is just called "action". This was a bit confusing as I thought that this was a different type of action.
- In the last line of "Resolving conflicts", "searching the CT in a best-first manner" - according to what value? the cost?
- In T-Robust CCBS, "However, if two agents do not collide when they are not delayed, there will not be any unsafe interval by CCBS." - I find this sentence very confusing. There are too many negations in this sentence.
- "TR-CCBS computes the unsafe interval of each action the other action." - I could not understand this sentence.
- "number of agents result in a denser environment" -> "number of agents results in a denser environment".
- In Table 1, the headline should be "Success Rate" and not "Success Ratio", as in other places.
- In "Comparison between CSG and Sampling", "different numbers of agents a=" - there are "n" agents and not "a" agents.

---

> ### Author Rebuttal · Authors · 2024-01-27
>
> 1) It is correct. However, changing the waiting time at some future vertex is an optimization technique of the plan execution stage, which is currently not considered in this paper. Our paper focuses on the plan generation stage: we aim to build a robust plan to tolerate a total delay up to T, without knowing when/where the delay will occur beforehand. The aforesaid execution-stage technique does not affect our definition of T-robustness: in the plan of an agent, any action can be the first action to suffer from a delay of T.
>
> 2) Yes, it is correct.
>
> 3) To clarify, TR-CCBS is solution complete, optimal, and sound. The solution completeness follows CCBS (Andreychuk et al. 2022) and CBS (Sharon et al. 2015). A solution complete algorithm is an algorithm that is guaranteed to find a solution if a solution exists. However, a solution complete algorithm may not detect unsolvability. That is, given an unsolvable problem, a solution complete algorithm may run indefinitely (Walker et al. 2020, see below).
> T. Walker, N.R. Sturtevant, A. Felner, Generalized and sub-optimal bipartite constraints for conflict-based search, in: The 24th AAAI Conference on Artificial Intelligence, AAAI, 2020, pp.7277–7284.
>
> 4) The plan cost in the lower part of Table 1 is for a particular problem (among the 250 problems tested), not the average plan cost of all the problems solved within the time limit. We did not use the average plan cost in Table 1, because the set of problems solved within the time limit is different for different K, T, and agent number settings (so it is not meaningful to compare the average plan cost across different settings).

---

### Meta-Review · Area_Chair_5Dh7 · 2024-02-05

**Recommendation:** Accept (Oral)
**Confidence:** 5

**Metareview:**

The paper studies T-robust MAPF, a combination of k-robust MAPF with continuous-time MAPF. It is well-written and well-organized. It proposes a CBS-based algorithm for solving T-robust MAPF optimally. While the proposed algorithm is a straightforward combination of the algorithms for k-robust MAPF and continuous-time MAPF, it studies the unique challenge posed by this combination, namely collision detection, in-depth and provides an exact method for solving it. The significance of the paper could be strengthened if it includes some strawman algorithms for empirical comparison.

**Ethical Considerations:**

(1) Not Applicable: The paper does not have any ethical considerations to address